# Perceived Provision of Perioperative Information and Care by Patients Who Have Undergone Surgery for Colorectal Cancer: A Cross-Sectional Study

**DOI:** 10.3390/ijerph192215249

**Published:** 2022-11-18

**Authors:** Alison Zucca, Elise Mansfield, Rob Sanson-Fisher, Rebecca Wyse, Sally-Anne Johnston, Kristy Fakes, Sancha Robinson, Stephen Smith

**Affiliations:** 1School of Medicine and Public Health, College of Health, Medicine and Wellbeing, University of Newcastle, Newcastle, NSW 2308, Australia; 2Priority Research Centre for Health Behaviour, University of Newcastle, Newcastle, NSW 2308, Australia; 3Department of Colorectal Surgery, Division of Surgery, John Hunter Hospital, New Lambton Heights, NSW 2305, Australia; 4Department of Anaesthesia, John Hunter Hospital, New Lambton Heights, NSW 2305, Australia; 5Department of Anaesthesia, Calvary Mater Newcastle Hospital, Newcastle, NSW 2298, Australia; 6Department of Surgery, Calvary Mater Newcastle Hospital, Newcastle, NSW 2298, Australia

**Keywords:** cancer, surgery, preparation for medical procedures, patient centered care, patient preferences, health care, colorectal, bowel

## Abstract

Background: Active patient participation in preparation and recovery from colorectal cancer surgery can be facilitated by timely information and care and may improve patient wellbeing and reduce hospitalizations; Methods: We aimed to identify gaps in perioperative information and care by asking colorectal cancer surgical patients to retrospectively report on their perceptions of care via a cross-sectional survey; Results: Overall, 179 (64% consent rate) patients completed one of two 64-item surveys exploring their views of ‘optimal care’ or their experiences of ‘actual care’. In total, 41 (64%) aspects of care were endorsed as optimal. Of these, almost three-quarters (73%) were received by most patients (80% or more). Gaps in care were identified from discrepancies in the endorsement of optimal versus actual survey items. Of the 41 items identified as representing ‘optimal care’, 11 items were received by fewer than 80% of patients, including the provision of information about the impact of surgical wait-times on cancer cure (69%); pre-habilitation behaviors to improve health (75%); the type of questions to ask the health care team (74%); impact of pain medications on bowel movements (73%); how to obtain medical supplies for self-care at home (67%); dietary or exercise advice after discharge (25–31%); and emotional advice after discharge (44%). Conclusions: These gaps represent patient-centered priorities and targets for supportive interventions.

## 1. Introduction

Colorectal cancer is the third highest incidence of all cancers worldwide, with surgical resection being the most common treatment received by at least 80% of patients in developed countries [1]. Surgery can result in numerous complications that impact the effectiveness of the surgery and overall patient wellbeing [2]. Enhanced Recovery After Surgery (ERAS) pathways are aimed at optimizing recovery following surgery and include evidence-based clinical care protocols under the direct control of the clinician (e.g., antibiotic prophylaxis) as well as patient-directed behaviors (e.g., early postoperative mobilization). Implementation of ERAS in colorectal cancer can reduce the length of hospital stay, post-surgery complications, and readmissions, and are cost-effective [3,4,5]. 

The active participation of the patient in the preparation and recovery from colorectal cancer surgery is vital across the entire perioperative period [6]. To physically and emotionally prepare for surgery, ERAS pathways recommend that patients eat nutritious food, exercise daily, quit smoking, cease alcohol consumption, become familiar with surgical and anesthetic procedures, and reflect on their expectations about the surgery and recovery process [6]. Post-operatively, commencing the day after surgery, ERAS pathways recommend patients can attempt to recommence eating and drinking, mobilization, respiratory physiotherapy, and balance opioid-based analgesia with the increased risk of constipation and slower bowel recovery [6]. After discharge, ERAS recommendations include patients actively seeking to manage side effects at home, recognizing and seeking care for emergency symptoms, and scheduling and attending follow-up appointments [6].

Research to date has established that in order to be an active participant in their preparation for and recovery from surgery, patients need timely information and care that addresses the aspects of their care that they perceive as important and that can build their understanding, knowledge, and confidence to work in partnership with their clinicians [7,8,9,10,11]. These principles are central to the concept of patient-centered care, and improve patient physical outcomes, health-related quality of life, satisfaction, knowledge, enable self-management, reduce hospitalizations, and improve cancer survival [12,13,14].

Examining patient-centered priorities for perioperative information and care can provide clarity on how to improve patient information and care [15,16,17,18,19]. To do this, there is value in understanding those components of their care that patients perceive as most important; and whether such components have been addressed by their healthcare team. Any resulting gaps between what patients perceive to be optimal care, and that which they receive, represent a gap in care delivery and a subsequent opportunity to assign priority for improvement [20]. Gaps can limit patients’ ability to undertake effective self-care before and after surgery, with the potential to affect clinical outcomes including length of stay and postoperative outcomes [10,21,22,23]. 

Consequently, this study aimed to identify gaps in perioperative information and care delivery among patients who have undergone surgery for colorectal cancer, by comparing patient perceptions of optimal care versus experiences of actual care received.

## 2. Materials and Methods

### 2.1. Design and Setting

This study is a cross-sectional study of patients who had undergone surgery to treat colorectal cancer. Participants retrospectively reported on their perceptions of perioperative care in one of two surveys investigating actual care received and care that is perceived as optimal. Eligible participants were within the past 18 months after receiving bowel resection surgery, and recruited in one of two large hospitals (public and private) in one metropolitan region of New South Wales (NSW) Australia. Together, these hospitals see over 50% of colorectal cancer surgical patients within the region. The composition of the treatment team, and the care provided is similar at both hospitals. Most of the surgeons work across both hospitals, however, nursing and allied health staff work exclusively in one hospital. Both hospitals provide patients with two pre-surgical appointments—first, the surgeon is consulted, and during the second appointment specialist colorectal nursing staff, and an anesthetist is consulted. Patients typically receive daily care from ward staff, and are also visited by a member of the surgical team, and physiotherapist. In the public hospital, a specialist colorectal cancer liaison nurse also visits each day. Access to other members of the allied health care team is provided as needed: including the stomal therapy nurse, dietician, and social worker. Patients are encouraged to visit their general practitioner within two weeks of discharge. At approximately one-month post-surgery, patients attend a follow-up consultation with the surgeon.

All eligible patients admitted to these two hospitals were invited to participate. Eligible patients completed one of two cross-sectional surveys depending on the time since surgery at sample selection. Optimal care items and actual care items were asked to separate samples of patients to reduce potential gratitude and social desirability bias [24] whereby patients reflecting on what was deemed optimal care may have influenced their responses to actual care items. To limit recall bias, patients who most recently underwent surgery (≤9 months ago) were retrospectively asked about their actual experiences of perioperative care, while those who had surgery 9–18 months ago were retrospectively asked for their views on what should be included as part of ‘optimal’ perioperative care. The 18-month post-surgery time frame was also a pragmatic choice to achieve an adequate sample size, which reflected patient throughput at the participating cancer clinics. To limit time as a confounding variable, patients were approached at one of two time points, so that both optimal and actual surveys captured responses from patients attending the services before and after the commencement of the COVID-19 pandemic.

### 2.2. Participants

Participants were eligible for the study if they had a confirmed diagnosis of colorectal cancer, were aged 18–80 years, and had undergone a bowel resection within the past 18 months. Data collection was conducted from April 2020 to September 2021.

### 2.3. Recruitment and Data Collection

Clinic staff identified eligible patients from a search of medical records. Patients were invited to participate in the survey via a mailed recruitment pack from their surgeon. The recruitment packs provided patients with the option of completing the survey either by an enclosed hard copy or via an online link. Reminder packs were sent after 4 weeks of non-response. Consenting patients returned their survey directly to the research team. The gender and age of all eligible patients were collected from medical records to assess response bias.

### 2.4. Measures

Optimal Care survey: A study-specific survey was developed as there are currently no patient-reported experience measures (PREMS) that explore patients’ views and observations about a comprehensive range of components of perioperative care delivery based on the Enhanced Recovery After Surgery (ERAS) pathway. The published literature was assessed to identify a comprehensive range of components of perioperative care delivery based on the (ERAS) pathway for colorectal cancer surgery [25], unmet patient needs [26], the Institute of Medicine’s key principles of patient-centered care [27], and a review existing instruments [24,28]. Items from existing instruments [29] were adapted to more accurately reflect the experience of colorectal cancer surgery, and extended to ask about the post-operative and post-discharge phases. For this study, an expert advisory group of researchers and clinicians (surgeon, anesthetist, colorectal cancer nurse, behavioral scientist) reviewed all items. Next, a purposive sample of 26 colorectal cancer patients who had undergone colorectal cancer surgery completed the draft items and identified their main concerns and perceived information needs throughout the perioperative period. Items were revised to reflect patient needs and reviewed again by the advisory group. This iterative consultation and input from both patients and expert advisory helped to refine and confirm the relevance of the items included in the instrument. The final survey consisted of 64 items in total. Items were divided into five phases reflecting the perioperative care journey as shown in Table 1.All items in the Optimal Care survey were preceded by the following question stem: ‘If a hospital is providing the best care possible, how important is it that the healthcare team…’? Participants responded on a five-point scale including the options ‘Not important’, ‘Slightly important’, ‘Moderately important’, ‘Very important’ or ‘Essential’.Actual Care survey: This survey comprised the same 64 items that were included in the Optimal Care survey, however, participants were asked ‘Did your healthcare team give you…. [information about…]/[offer advice or referral about]…’? Participants responded with either ‘Yes’ or ’No’. For ten items that may not have been applicable to all patients (e.g., smoking, alcohol, drug cessation; return to work or driving; chronic pain, or anxiety), a ‘Not applicable’ response option was provided.Sociodemographic, disease, and treatment variables: Each survey included the following items: patient age; gender; education; employment status; living arrangements; time since diagnosis; time since surgery; travel time to treatment.

### 2.5. Analysis

Possible response bias for participant age and gender was tested using the chi-square test for independence. To identify gaps in perioperative care, data from the optimal and the actual survey were compared. First, items from the Optimal Care survey that were rated as either ‘Very important’ or ‘Essential’ by at least 80% of participants were identified and considered to be ‘endorsed as optimal’. A cut point of 80% reflects a standard approach for measuring consensus [30]. Next, for each of these items, the proportion of patients who indicated in the Actual Care survey that they received the specified information and care was calculated. Consensus between the participating services and the advisory group deemed one in five patients (20%) or more not having received an aspect of preoperative care as an important ‘red flag’ service delivery issue for further investigation. A ‘gap’ was defined as existing when an aspect of information and care deemed important by 80% or more of patients (i.e., optimal component care), but was received by 80% or fewer patients. Missing items were treated as missing and removed from the denominator for calculations.

## 3. Results

A total of 282 eligible patients were identified through a medical record search and invited to participate. Of these, 179 (63.5%) returned a completed survey (*n* = 79, 61.7% ‘Optimal care’ survey; *n* = 100, 64.9% ‘Actual care’ survey). There were no significant differences between consenters compared to non-consenters in terms of gender (*χ*^2^ (1) = 0.242, *p* = 0.623). However, compared to non-consenters ($\bar{x}$ = 65.8 years, SD = 12.3), consenters ($\bar{x}$ = 70.6 years, SD = 10.1) were of older age at the time of surgery (t(278) = 3.49, *p* = 0.0006). Table 2 shows participant characteristics. The total sample was generally representative of those Australians receiving surgery for colorectal cancer in terms of gender (55% male) and age (75–79 years most likely to receive surgery) [31]. There were no significant differences in participant characteristics between samples completing the ‘actual’ and ‘optimal’ versions of the survey (see Table 2).

Overall, 41 (64%) of the 64 aspects of care presented to participants were endorsed as optimal (see Table 3). Of these, 30 (73%) were received by most (80% or more) patients. Table 4 lists the remaining 23 items not endorsed as optimal.

At the pre-hospital phase, of the 16 items endorsed as optimal, 12 were received by most (80% or more) patients. The optimal aspects of care most commonly received were information and care relating to what needs to happen before the procedure (97%), where to go when arriving at the hospital (97%), and possible risks or complications (94%). Four gaps in care were identified in this phase: provision of information about other treatment options available (51% of patients did not receive any information); whether the waiting time for surgery will impact on cure (31% of patients did not receive any information); the type of questions to ask (26% of patients did not receive any information) and things to do to improve health (25% of patients did not receive any information).

At the pre-operative phase, of the five items endorsed as optimal, all were received by most (80% or more) of patients. Information regarding what will occur if something unexpected happens during surgery was least commonly received, with 18% of patients not having received any information.

At the post-operative phase, of the four items endorsed as optimal, three were received by most (80% or more) patients. Those aspects of care most commonly received were the importance of early mobilization (88% received), information about emergency symptoms (85% received), and deep breathing and coughing (84% received). One gap was identified: information about how pain medications affect bowel movements (27% of patients did not receive any information).

At the discharge planning phase, of the 12 items endorsed as optimal, 10 aspects of care were received by most (80% or more) patients. Those aspects of care most commonly received were information about the necessary follow-up appointments (97% received), whom to contact for further advice (94%), whom to contact for about urgent symptoms (94%), and the type of symptoms to urgently seek care for (92%). Two gaps were identified: information about how to obtain medical supplies (33% of patients did not receive any information) and dietary information before leaving the hospital (23% did not receive any information).

At the post-discharge follow-up phase, of the four items endorsed as optimal, all were identified as gaps. The largest gap was addressing feelings of distress, worry, or sadness. Of the 45 (59%) patients who identified distress, worry, or sadness as an issue applicable to them, 25 (56%) did not receive any advice or referral from their healthcare team. Of the 52 (68%) patients for whom chronic pain was applicable, 50% did not receive any advice or referral from their healthcare team. Also, almost one-third of patients (31%) reported not receiving diet and nutrition advice or referral, and one-quarter (25%) did not receive advice about exercise after discharge.

## 4. Discussion

This study presents colorectal cancer patient perceptions of optimal perioperative information provision and support and gaps in care delivery. Overall, high-quality care was reported by most patients, particularly regarding the provision of information about the surgery and pre-surgical procedures. However, consistent with previous research, several gaps in information and care were identified pre-admission, post-operatively, at discharge, and during follow-up post-discharge [15,16,17,18,19,32]. These gaps suggest that not all patients were being instructed about pre- and re-habilitation behaviors (diet, exercise, and symptom management), nor offered emotional support from their health care team. These gaps present a priority for service improvement, as they can affect patient self-care, and in-turn length of stay and postoperative mortality [10,22,23].

### 4.1. Address the Impact of Surgical Wait Times and Pre-Habilitation Behaviors

Almost one-third of patients did not receive information about whether surgical wait-time would impact cancer cure. Concerns about cancer quickly spreading and the urgency to have surgery are a key source of patient distress [33,34]. In Australia, most of the delay for treatment occurs at the diagnosis phase (52 days) rather than the interval after a positive diagnosis to surgery (30 days) [35,36,37]. While there is increased morbidity and mortality associated with a longer wait time to diagnosis, once diagnosed, there is no evidence that a longer surgical wait-time is associated with poorer morbidity and survival outcomes [37,38,39]. Surgical services could consider routinely including this evidence in the information that is provided to patients. Furthermore, the period of time between diagnosis and surgery could be used as an opportunity to emphasize those beneficial pre-operative behaviors that can improve recovery outcomes, and which are under the direct control of patients [40]. Activating control can help to address feelings of helplessness and distress during this waiting period [40].

### 4.2. Nutrition Information and Care

Consistent with previous research, gaps in information and care for eating and nutrition were identified pre-operatively, at discharge, and follow-up [16,19,37]. Rapid resumption of normal eating after surgery is complicated by ileus or gastrointestinal stasis, lack of appetite, food intolerance, adapting the diet to having a stoma, and dissatisfaction with hospital food [16,19,41]. Also, dietary challenges continue to be a long-term challenge for patients who report an altered relationship with food [16]. Colorectal surgical services could consider providing additional dietary information and professional support to all patients—such as listing foods that commonly cause symptoms; or practical strategies such as keeping a food diary to record and monitor the effect of different foods on the bowel [42].

### 4.3. Facilitating Question Asking

During the pre-admission phase, some patients reported not receiving advice about the type of questions they should ask their healthcare providers. A structured list of questions (question prompt lists, ‘QPLs’) are communication aids to encourage active patient participation in health care discussions. They can be used alongside lists of frequently asked questions to enhance patient education. QPLs are acceptable to patients, can increase the total number of questions asked, prompt necessary and difficult discussions, reduce anxiety, and shorten the length of consultations [43,44,45,46,47]. QPLs may help support patient engagement and communication with healthcare providers and may be a cost-effective and helpful way to improve care delivery when used in routine practice [43,45,46,47].

### 4.4. Pain and Symptom Education and Support

While opioids are essential to relieve patient suffering and enable mobilization after surgery, they cause constipation, a symptom experienced by 40–95% of colorectal cancer surgical patients [48]. In this study, almost one-quarter of all patients were not provided with any information regarding the interaction between pain medications and constipation, and so were unable to make informed choices about minimizing opioids [48]. At the follow-up appointment, less than half of respondents were offered advice or referral for their chronic pain, a common symptom experienced by up to 40% of colorectal cancer survivors 4 years after treatment [49,50]. Chronic pain may require further in-depth specialist follow-up. Services should consider providing information about chronic pain to patients as standard care.

### 4.5. Medical Supplies

At the time of discharge, more than one-third of patients reported a gap in education and care about how to obtain general medical supplies such as dressings for wounds, barrier creams, perineal washcloths to prevent sore skin, absorbent underwear, and soaps. However, fewer patients reported issues accessing stoma care services. With proper use of medical supplies, patients can be well-prepared to prevent or manage side effects. Messaging should be clear and specific and include which products the patient is responsible for buying, where and what to buy, the quantity of products needed, and detailed instructions about how to use these products.

### 4.6. Emotional Support

During post-discharge follow-up appointments, more than half of patients who had feelings of distress, worry, or sadness were not offered advice or referral by their healthcare team. Gaps in psychosocial care for colorectal cancer patients have been well documented [49]. These gaps may reflect time constraints in provider knowledge, skills, and self-efficacy for handling psychosocial concerns or that some providers may await for their patients to explicitly ask for emotional support [51,52]. Clinical pathways suggest triaging patients by first addressing concerns and acknowledging the patients’ distress, offering all patients in need with access to resources (e.g., web-based), promoting social support, and offering professional support from mental health services [53].

### 4.7. Exercise Education and Support after Discharge

At post-discharge follow-up appointments, almost one-quarter of colorectal cancer patients did not receive advice about exercise from their healthcare team. While patient physical activity levels and exercise tolerance is recorded at pre-admission, this information is only used by surgical services as a proxy for cardiorespiratory fitness for their surgery, rather than to assess and encourage routine exercise. A recent meta-analysis recommends 300 min of moderate-intensity aerobic exercise per week, which can improve cardiopulmonary fitness, metabolism, tumor-related biomarkers, quality of life, and fatigue levels in post-treatment colorectal cancer survivors [54]. Post-treatment consultations should involve encouraging patients to become more physically active, or maintain sufficient activity, using evidence-based techniques [55,56].

### 4.8. Implications for Further Research/Practice

Given the constraints on the health system and clinician time, alternative ways of providing patients with this information and care should be investigated. Specifically, digital health interventions may hold promise in being an effective and cost-effective mechanism for providing such information and care. There is already some evidence from non-randomized studies suggesting that digital health interventions (DHI) can support colorectal cancer patients to better adhere to the patient-managed ERAS components [57]. Some of the gaps identified could be met via the provision of generic information (via written, website, or audiovisual), whereas, for others, a more tailored approach may be necessary. For example, the provision of generic information might be best suited to gaps about where to buy medical supplies, the provision of QPLs, and advice about pain medication and constipation. However, a more tailored approach might be suited to nutritional information (stoma versus no-stoma), exercise education (open surgery versus laparoscopy), and emotional support (distress versus no distress) where the patient’s individual circumstances may mean that generic advice is not appropriate. Given DHIs can be dynamic, flexible, and tailored to the needs of the individual patient, they represent efficient ways of providing information and care and may be useful to address existing gaps within the health system. The research team are currently trailing such an intervention [58], which (in addition to supporting adherence to the ERAS recommendations) specifically addresses the gaps: by emphasizing beneficial pre-operative behaviors including healthy diet and exercise; how pain medications affect bowel movements; how to access medical supplies; nutrition information and care across the perioperative period. 

Future research should continue to collect and summarise qualitative data to describe patient healthcare experiences and their unmet needs, and use these data to improve existing quantitative measures. Also, this research identified gaps in care delivery that were found across two sites. We have not explored the predictors of these gaps. Given that individual patient and treatment center factors have been found to impact care delivery, future multi-site studies could explore organizational characteristics that impact care such as funding source, teaching status of the hospital, number of staff, number of disciplines represented among staff, policies, and procedures surrounding perioperative care, clinician-patient continuity of care, hospital size, and/or the team culture of the clinic. Finally, a systematic review of patient-reported experience measures (PREMS) for colorectal surgery perioperative care would make an important contribution to the literature.

### 4.9. Strengths and Limitations

The patients were recruited from two hospitals in one region of Australia representing over 50% of colorectal cancer surgical patients in the region and captured the diversity of care delivery for both public and private patients. The disease and demographic characteristics of this study sample are broadly representative of Australian colorectal surgical patients more generally [31], and thus it is likely that the findings are somewhat generalizable to patients across Australia. However, given that consenters were five years older than non-consenters on average, we may have underestimated the number and extent of gaps, as older patients report fewer unmet needs [59]. Also, the gaps in care identified may differ from hospitals in other regions or countries. Also, while this study focused on ‘red flags’, there were other aspects of care not received regularly by patients that may also require further investigation. A study-specific survey was developed as no previous survey was identified that examined the comprehensive perioperative pathway from pre-hospital to post-discharge. However, this adapted version of the survey has not undergone further psychometric testing. We asked patients to retrospectively report on their perceptions of care using a cross-sectional survey. To reduce recall bias, those who reported on their actual surgical experiences were within nine months of surgery. However, it is possible that this time frame may have still resulted in some recall bias. Asking patients about care expectations and delivery during the cancer journey via a prospective longitudinal survey may have resulted in a different perspective. Also, we chose to explore the functioning of the surgical services unit rather than exploring the actions of individual healthcare professionals. Although our data do not allow feedback to be provided to specific health professional groups, they reflect care delivery within organizational units of multidisciplinary teams. Also, gaps in care were derived using between-person differences based on separate samples, rather than within-person differences, which may be a less sensitive approach. This approach was taken to reduce participant burden, and also to reduce gratitude and social desirability bias—that is, patients are often reluctant to report dissatisfaction [24], so by reflecting on what was deemed optimal care this may have influenced their responses to actual care items. We did not collect any data about the type of bowel resection (laparoscopy vs. open) nor the patient stoma status after surgery, so we could not explore whether gaps differed by these sub-groups. Finally, many of these patients may had their surgery during COVID and this may not have always reflected typical care during non-pandemic times.

## 5. Conclusions

This study highlights key gaps in the provision of perioperative education and care identified through the eyes of the patient. Gaps include information about the impact of surgical wait-times on cancer cure, other treatment options available, pre-habilitation behaviors to improve health, the type of questions to ask during consultations, impact of pain medications on bowel movements; how to obtain medical supplies for self-care activities at home; dietary or exercise support at discharge and follow-up; and emotional support at follow-up. These patient-centered priorities for information and care could be considered in conjunction with best-practice guidelines to improve perioperative care and support. Equipping all patients with self-management knowledge may help to promote adherence to patient-directed behaviors on the ERAS pathways, improve patient wellbeing and reduce hospitalizations.

## Figures and Tables

**Table 1 ijerph-19-15249-t001:** Summary of survey domains and items by phase of perioperative care.

Phase of Perioperative Care	Domain	Number of Items	Optimal Care Example	Actual Care Example
Pre-hospital	Preparation for bowel cancer surgery	24	If a hospital is providing the best care possible, how important is it that the healthcare team gives patients information before their bowel cancer surgery about: the things they can do to improve their health while waiting for surgery	Before having bowel cancer surgery, did your healthcare team give you information about: The things you can do to improve your health while waiting for surgery
Pre-operative	Having bowel cancer surgery	8	If a hospital is providing the best care possible, how important is it that the healthcare team gives patients information about: What will happen during the procedure	Did your healthcare team give you information about: What will happen during the procedure
Post-operative	Hospital care immediately after bowel cancer surgery	7	If a hospital is providing the best care possible, how important is it that the healthcare team gives patients information about: When it is possible to start eating again	Did your healthcare team give you information about: When it is possible to start eating again
Discharge planning	Preparation for discharge from hospital	18	If a hospital is providing the best care possible, how important is it that the healthcare team gives patients information before they leave hospital about: Management of side effects	Before you left hospital, did your healthcare team give you information about: Management of side effects
Post-discharge follow-up	Post-surgery follow-up appointments	7	If a hospital is providing the best care possible, how important is it that the healthcare team offers advice or referral (if needed) following surgery for: chronic pain issues	Following your surgery, did your healthcare team offer advice or referral (if needed) for chronic pain

**Table 2 ijerph-19-15249-t002:** Demographic, disease and treatment characteristics of participants.

Variable	*N*	Category	Optimal Care (*n* = 79)	Actual Care (*n* = 100)	Total (*n* = 179)		*p* Value
Age at survey completion	178	Mean (SD)	71.4 (10.4)	70.0 (10.22)	70.7	(10.33)	0.363
Travel time from home to hospital for surgery (minutes)	178	Mean (SD)	66.8 (79.8)	66.5 (150.4)	66.3	(123.8)	0.982
					*N*	(%)	
Gender	178	Male	53%	60%	101	(56%)	0.390
Education attained	178	School certificate or higher school certificate	51%	42%	97	(55%)	0.460
Current employment status	177	Retired or age pension	80%	69%	130	(73%)	0.105
		In paid employment	14%	15%	26	(15%)	
		Not in paid employment, disability pension, sick leave	6%	16%	21	(12%)	
Living arrangements	177	Living with other/s (versus alone)	71%	72%	127	(72%)	0.818
		Living alone	29%	28%	50	(28%)	
Hospital type	176	Public	59%	62%	107	(61%)	0.750
		Private	41%	38%	69	(39%)	
Time since diagnosis at survey completion	175	9 months ago or less	0	63%	62	(35%)	N/A ^
		9 to 12 months ago	18%	37%	50	(29%)	
		more than 12 months ago	82%	0	63	(36%)	
Time since surgery	178	9 months ago or less	0	80%	79	(44%)	N/A ^
		9 to 12 months ago	35%	20%	48	(27%)	
		more than 12 months ago	65%	0	51	(29%)	

*n* does not add up to *N* because of missing data; ^ To limit recall bias, those most recently undergoing surgery completed the ‘Actual care’ survey.

**Table 3 ijerph-19-15249-t003:** Aspects of perioperative information and care endorsed as ‘optimal’ by 80% or more of patients: perceptions of optimal care versus experiences of actual care received.

Phase of Perioperative Care Aspects of Perioperative Information and Care Delivery	Endorsed as Optimal Perioperative Care *n* = 79 * *n* (%)	Actual Care Received *n* = 100 * *n* (%)	Gap in Perioperative Care Delivery ^
Pre-hospital: Preparation for bowel cancer surgery			
What could happen if they didn’t have surgery	73 (97.3%)	80 (81.6%)	
What needs to happen before procedure	72 (94.7%)	97 (97.0%)	
Who to contact if they have any questions	70 (94.6%)	93 (93.0%)	
Talk about any fears or worries with healthcare team	72 (92.3%)	84 (85.7%)	
Provide information in the amount of detail preferred	71 (91.0%)	86 (88.6%)	
The expected benefits	69 (90.8%)	88 (88.9%)	
The importance of stopping smoking before surgery	68 (89.5%)	13 (86.7%)	
Possible risks or complications	67 (89.3%)	94 (94.0%)	
Likely care needed from family or friends after surgery	67 (85.9%)	86 (86.0%)	
Aside from surgeon, the health professionals likely to see	63 (82.8%)	91 (92.8%)	
Provide information in preferred format(s)	62 (82.7%)	85 (86.7%)	
Where to go when arriving at hospital	62 (82.7%)	97 (97.0%)	
Any other treatment options available	61 (80.3%)	48 (48.9%)	Gap
Whether the waiting time for surgery will impact on cure	66 (88.0%)	66 (68.7%)	Gap
The type of questions they should ask(e.g is it possible to cure or control my cancer?; how much will the treatment or test cost?; are there any clinical trials suitable for me?)	68 (89.5%)	71 (73.9%)	Gap
Things they can do to improve their health	66 (88.0%)	71 (74.7%)	Gap
Pre-operative: Having bowel cancer surgery			
What will occur if something unexpected happens during surgery	71 (92.2%)	81 (81.8%)	
Getting pain relief after surgery	70 (90.9%)	91 (91.9%)	
Preparations just before going into surgery	66 (85.7%)	95 (97.9%)	
What they might feel immediately after waking up from surgery	66 (85.7%)	86 (86.9%)	
What will happen during the procedure	64 (84.2%)	90 (90.9%)	
Post-operative: Hospital care immediately after bowel cancer surgery			
Symptoms requiring immediate attention of hospital staff	74 (97.4%)	82 (84.5%)	
The importance of deep breathing and coughing in aiding recovery	69 (90.8%)	82 (83.7%)	
The importance of getting out of bed and moving on the day of surgery	67 (88.1%)	87 (87.8%)	
How pain medications affect bowel movements	65 (85.5%)	72 (72.7%)	Gap
Discharge planning: Preparation for discharge from hospital			
Symptoms or side-effects they should urgently seek care for	76 (98.7%)	88 (91.7%)	
How to manage any side-effects or complications	75 (96.1%)	87 (88.8%)	
Who to contact about urgent symptoms and side-effects	75 (96.1%)	91 (93.8%)	
How to care for their wounds	73 (96.0%)	89 (89.9%)	
A written plan for their care after discharge (a document describing any ongoing medication, future tests, follow-up appointments etc.)	72 (96.0%)	81 (82.6%)	
Who to contact for further advice, information, questions and concerns	74 (95.5%)	93 (93.9%)	
Follow-up appointments to make with GP and/or specialist	69 (89.6%)	96 (97.0%)	
How to take medications for post-surgery recovery correctly	69 (89.6%)	79 (82.3%)	
How to access stoma care services ^#^	63 (88.7%)	38 (88.4%)	
Exercise and activity	65 (84.4%)	83 (84.7%)	
How to obtain medical supplies—such as wounds dressings; cleansing products and creams for sore skin, absorbent underwear—provision of information before leaving hospital	62 (84.9%)	62 (67.4%)	Gap
Eating and nutrition	65 (84.4%)	75 (77.3%)	Gap
Post discharge follow-up: Post-surgery follow-up appointmentsOffered advice or referral if needed for:			
Chronic pain ^#^	70 (92.1%)	26 (50.0%)	Gap
Exercise	69 (90.8%)	74 (74.7%)	Gap
Diet and nutrition	67 (89.3%)	68 (69.4%)	Gap
Feelings of distress, worry or sadness ^#^	66 (86.8%)	20 (44.4%)	Gap

**^^^** Endorsed as optimal (rated as important by 80% or more of patients) but received by less than 80% of patients. ***** Missing data: 96% of actual care survey participants had provided data for all actual care items, and 96% of optimal participants provided data for all care items. ^#^ Item included a “not applicable” respond option (stoma: 54 responded not applicable; chonic pain: 48 responded not applicable; feelings of distress: 55 responded not applicable).

**Table 4 ijerph-19-15249-t004:** Aspects of perioperative information and care endorsed ‘optimal’ by fewer than 80% of patients (*n* = 23 items).

Phase of Perioperative Care Aspects of Perioperative Information and Care Delivery	Endorsed as Optimal Perioperative Care *n* = 79 * *n* (%)	Actual Care Received *n* = 100 * *n* (%)
Pre-hospital: Preparation for bowel cancer surgery		
What to take to hospital	51 (67.1%)	83 (83.8%)
Car parking and transport to and from hospital	51 (67.1%)	80 (81.6%)
What ward they will be in after surgery	48 (64%)	74 (74%)
How family and friends can contact hospital	56 (71.8%)	82 (82%)
Hospital visiting rules	46 (59.7%)	77 (80.2%)
Likely length of hospital stay	55 (70.5%)	97 (97%)
Likely time off work to recover	53 (70.7%)	27 (93.1%)
About the experiences of other people	36 (46.7%)	41 (42.7%)
Pre-operative: Having bowel cancer surgery		
Strategies to help them manage any anxiety or stress before the procedure	52 (68.4%)	49 (50%)
Where they will wake up after surgery	52 (67.5%)	88 (89.8%)
How long they will spend in the recovery area following surgery	50 (64.9%)	79 (79.8%)
Post-operative: Hospital care immediately after bowel cancer surgery		
Expected activities on each day following the surgery	48 (63.1%)	77 (77.8%)
When it is possible to start eating again	48 (63.1%)	83 (83.8%)
New medications they will be taking	60 (78.9%)	62 (63.3%)
Discharge planning: Preparation for discharge from hospital		
When they should be able to return to work	46 (61.3%)	25 (92.6%)
When they should be able to return to driving	51 (66.2%)	68 (88.3%)
When they should be able to return to their usual activities	53 (68.8%)	82 (82.8%)
Additional trustworthy information resources about their cancer and treatment	60 (77.9%)	76 (80%)
How to access support services and support groups within the local community	55 (71.4%)	61 (64.9%)
How to manage their moods and emotions	55 (73.3%)	38 (40%)
Post discharge follow-up: Post-surgery follow-up appointments		
Offered advice or referral if needed for:		
Smoking ^#^	55 (74.3%)	8 (72.7%)
Recreational drug use ^#^	54 (74%)	2 (33.3%)
Alcohol ^#^	53 (70.7%)	18 (72%)

* Missing data was 96% of actual care survey participants had provided data for all actual care items, and 96% of optimal participants provided data for all care items. ^#^ Item included a “not applicable” respond option (smoking: 88 responded not applicable; recreational drug use: 92 responded not applicable; alcohol: 73 responded not applicable).

## Data Availability

The datasets used and/or analyzed during the current study are available from the corresponding author on reasonable request.

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
