# Peer review of "Perceived Provision of Perioperative Information and Care by Patients Who Have Undergone Surgery for Colorectal Cancer: A Cross-Sectional Study"

_ijerph, 2022, doi:10.3390/ijerph192215249_

Round 1

Reviewer 1 Report

1. I would suggest to the authors to justify in the Design and setting section why they chose a period of less than 9 months for the survey about actual expierence and to explain this sentence in more detail: "To limit recall bias, patients who most recently underwent surgery (≤9 months ago) were retrospectively asked about their actual experiences of perioperative care, while those who had surgery 9-18 months ago were retrospectively asked for their views on what should be included as part of ‘optimal’ perioperative care."

2. The title of Table 1 needs to be corrected: "Table 1. This is a table. Tables should be placed in the main text near to the first time they are cited."

3. The authors could write a little more about how the gaps found could be filled, elaborating on this sentence: "Some of the gaps identified could be met via the provision of generic information (via written, website, audio visual), whereas others, a more tailored approach may be necessary."

3. I would suggest clarifying what is meant by asking this question in the survey: "The type of questions they should ask".

Author Response

Thank you very much for the opportunity to submit revisions to Manuscript ID ijerph-2004228 entitled " Perceived provision of perioperative information and care by patients who have undergone surgery for colorectal cancer: a cross-sectional survey.  Thank you to the reviewers for their thoughtful responses. Please find our detailed responses to these reviewers’ comments below.

Reviewer 1:

  1. I would suggest to the authors to justify in the Design and setting section why they chose a period of less than 9 months for the survey about actual experience and to explain this sentence in more detail: "To limit recall bias, patients who most recently underwent surgery (≤9 months ago) were retrospectively asked about their actual experiences of perioperative care, while those who had surgery 9-18 months ago were retrospectively asked for their views on what should be included as part of ‘optimal’ perioperative care."

The following details have been added at line 108:

The 18-month post-surgery time-frame was also a pragmatic choice to achieve an adequate sample size which reflected patient throughput at the participating cancer clinics.

We have also acknowledged potential recall bias in the strengths and limitations section of the discussion

  1. The title of Table 1 needs to be corrected:

Table 1 title has been added.

  1. The authors could write a little more about how the gaps found could be filled, elaborating on this sentence: "Some of the gaps identified could be met via the provision of generic information (via written, website, audio visual), whereas others, a more tailored approach may be necessary."

The following details have been added at line 348:

For example, provision of generic information might be best suited to gaps about where to buy medical supplies, the provision of QPLs, and advice about pain medication and constipation. However, a more tailored approach might be suited to nutrition information (stoma versus no-stoma), exercise education (open surgery versus laparoscopy), and emotional support (distress versus no distress) where the patients’ individual circumstances may mean that generic advice is not appropriate.

  1. I would suggest clarifying what is meant by asking this question in the survey: "The type of questions they should ask".

Table 3 has been amended to include three examples typical of those included in a ‘question prompt list’ (QPL).

e.g is it possible to cure or control my cancer?; how much will the treatment or test cost?; are there any clinical trials suitable for me?

We hope our revised manuscript meets your approval.

Reviewer 2 Report

Thank you for getting the opportunity to review this very interesting, relevant and thoroughly executed study. In general, the article is well written and systematic presented. My more detailed comments are:

Line 87: This sentence does not make sense to me: “Optimal care items and actual items were asked to separate samples of patients…”

113 -143: You describe that there is 64 items in each questionnaire, but Table 1 describes only 53 items. Does 11 items concern demographic data or ?

Line 142: There is a missing title to Table 1.

Line 175: There is a missing title to Table 2.

Line 148: I wonder why p-values were not analyzed between the two groups for all data and not just for demographic data?

Line 222 (Table 3): There is a mismatch between the described number of items in Table 1 and the number of items presented in Table 3.  It would be beneficial if you refer to the Appendix here, so the reader do not not have to search for the “missing” results.

Line 223 – 316: The authors discuss many findings, which makes the discussing of some subjects a bit superficial.

Author Response

Thank you very much for the opportunity to submit revisions to Manuscript ID ijerph-2004228 entitled " Perceived provision of perioperative information and care by patients who have undergone surgery for colorectal cancer: a cross-sectional survey.  Thank you to the reviewers for their thoughtful responses. Please find our detailed responses to these reviewers’ comments below.

Reviewer 2

  1. Comments: Thank you for getting the opportunity to review this very interesting, relevant and thoroughly executed study. In general, the article is well written and systematic presented. My more detailed comments are:

  1. Line 87: This sentence does not make sense to me: “Optimal care items and actual items were asked to separate samples of patients…”

This sentence has been amended to improve comprehension.

  1. You describe that there is 64 items in each questionnaire, but Table 1 describes only 53 items. Does 11 items concern demographic data or ?

Table 1 included a typographical error that has been corrected. There are indeed 64 items in each questionnaire

  1. There is a missing title to Table 1.

The title has been added at line 162.

  1. There is a missing title to Table 2.

The title has been added

  1. I wonder why p-values were not analyzed between the two groups for all data and not just for demographic data?

Thank you. We did consider this option, but given the patient-centred approach of the study the study team deemed it more important to focus on the aspects of care that were important to patients, and whether actual care deviated from this. For example, a significant p value could occur when a service was well provided, but not necessarily important to patients.

  1. Line 222 (Table 3): There is a mismatch between the described number of items in Table 1 and the number of items presented in Table 3.  It would be beneficial if you refer to the Appendix here, so the reader do not not have to search for the “missing” results.

Table 3 has been formatted to improve comprehension. There are 41 items in Table 3, and 23 items in Appendix A1, equaling 64 items in total.  Appendix A1 has been moved to the main body of paper and is now Table 4.

  1. Line 223 – 316: The authors discuss many findings, which makes the discussing of some subjects a bit superficial.

This paper takes a holistic view of the provision of perioperative education and care by reporting on a number of components of care across each phase of the surgical journey.  A comprehensive discussion was difficult given the large number of care components that were asked about in the survey. In the manuscript, we have attempted to consider the wide range of issues here, based on the identified gaps, rather than focusing on 1-2 gaps only.

We hope our revised manuscript meets your approval.

Reviewer 3 Report

I find that the study design could be substantially improved.

The title "Provision of information" I believe that the title does not fit the results or initial objective, it is more focused on a study to identify the perceived quality of care of patients undergoing colorectal surgery.

The study does not specify if this project has passed through a clinical ethics committee and if it has been approved to carry it out.

The study tells us about a questionnaire but does not specify or make it very clear what the degree of feasibility and validity has been for the experts, and if these experts represent all areas of the researchers and the care that the patient must receive before and after treatment. post intervention.

It has been difficult for me to follow the study, and sometimes to identify according to the text what type of study, and its objective is lost, since it is a retrospective cross-sectional study, to identify the perioperative gaps and their care (and these by whom they should have done it), but it does not specify the different stages that are in the intervention process, and if these change or could change whether it is a public or a private one.

Regarding the variables, he talks about the sociodemographic ones but not the socioeconomic ones, since both pre and post care could vary in terms of the information perceived if it is a private or public hospital and which professionals intervene in one or the other. . don't compare it.

The study presents important gaps and gaps both in the design of the study and in the methodology, thus it is reflected in the discussion elaborated by the authors, therefore it is to be understood that this does not present significant differences in the characteristics of the participants of the study. study.

From the scientific point of view, it is more a study to identify the perceived quality of care of patients undergoing colorectal surgery, to look at the level of anxiety, the perceived care and the quality of these, to review there are validated questionnaires, which allow us to evaluate this perception, and the latest systematic reviews of it.

For future studies, they could assess qualitative studies that allow us to identify what these perceived needs by patients could be, and accompany it with quantitative studies as they present us, improving the variables that are intended to be investigated, the tools that are used, and better justify why, and what is intended to improve, in this case the gaps in information and the quality of care perceived by the patient.

I think you should do a review again and make a new article proposal, which improves the quality and viability of cross-sectional studies and of interest to readers, and clarifies the stages and professionals who have to carry out the interventions, let us know if it has happened an ethics committee, and better clarify the viability of the survey carried out and its dimensions in accordance with the objective or objectives set out in its study.

Author Response

Thank you very much for the opportunity to submit revisions to Manuscript ID ijerph-2004228 entitled " Perceived provision of perioperative information and care by patients who have undergone surgery for colorectal cancer: a cross-sectional survey.  Thank you to the reviewers for their thoughtful responses. Please find our detailed responses to these reviewers’ comments below.

Reviewer 3

  1. Comments: I find that the study design could be substantially improved.
  2. The title "Provision of information" I believe that the title does not fit the results or initial objective, it is more focused on a study to identify the perceived quality of care of patients undergoing colorectal surgery.

The title has been updated to reflect that this cross-sectional survey descries patient’s perception of perioperative information and care

  1. The study does not specify if this project has passed through a clinical ethics committee and if it has been approved to carry it out.

The Institutional Review Board Statement on Line 422 included these details.

  1. The study tells us about a questionnaire but does not specify or make it very clear what the degree of feasibility and validity has been for the experts, and if these experts represent all areas of the researchers and the care that the patient must receive before and after treatment. post intervention.

Line 130 to 148 describes development of the study specific survey, including the research and clinical qualifications of the advisory group who developed the items.

It has been difficult for me to follow the study, and sometimes to identify according to the text what type of study, and its objective is lost, since it is a retrospective cross-sectional study, to identify the perioperative gaps and their care (and these by whom they should have done it), but it does not specify the different stages that are in the intervention process, and if these change or could change whether it is a public or a private one.

  • The title and methods section (line 80) has been revised to better reflect the nature of the study as cross-sectional, and exploring patient perceptions.
  • More detail about the trajectory of care and composition of the perioperative team at each hospital is provided (added line 87).

The composition of the treatment team, and the care provided is similar at both hospital.  Most of the surgeons work across both hospitals, however nursing and allied health staff work exclusively in one hospital. Both hospitals provide patients with two pre-surgical appointments – first the surgeon is consulted, and during the second appointment specialist colorectal nursing staff, and an anesthetist is consulted. Patients typically receive daily care from ward staff, and are also visited by a member of the surgical team, and physiotherapist. In the public hospital, a specialist colorectal cancer liaison nurse also visits each day. Access to other members of the allied health care team is provided as needed: including stomal therapy nurse, dietician, social worker. Patients are encouraged to visit their general practitioner within two weeks of discharge. At approximately one-month post-surgery, patients attend a follow-up consultation with the surgeon.

  • Added to line 388:

Also, we chose to explore the functioning of the surgical services unit rather than exploring the actions of individual health professionals. Although our data do not allow feedback to be provided to specific health professional groups, they reflect care delivery within organizational units of multidisciplinary teams”

  1. Regarding the variables, he talks about the sociodemographic ones but not the socioeconomic ones, since both pre and post care could vary in terms of the information perceived if it is a private or public hospital and which professionals intervene in one or the other. . don't compare it.

 While we have reported patients current employment status and education attained, which are proxy measures of social-economic status, it is beyond the scope of this manuscript to explore the predictors of gaps in care delivery. However, we have discussed this in more detail (from line 362)

This research identified gaps in care delivery that were found across two sites. We have not explored the predictors of gaps. Given that individual patient and treatment centre factors have been found to impact care delivery, future studies could consider exploring organizational characteristics that impact on care including funding source, teaching status of the hospital, number of staff, number of disciplines represented among staff, policies and procedures surrounding perioperative care, clinician-patient continuity of care, hospital size, and/or the team culture of the clinic.

For more context, we have described the surgical patients typical journey, and typical interaction with health care professionals at each stage of the cancer surgical journey.

  1. The study presents important gaps and gaps both in the design of the study and in the methodology, thus it is reflected in the discussion elaborated by the authors, therefore it is to be understood that this does not present significant differences in the characteristics of the participants of the study. study.

We were not quite sure of the meaning of this particular comment. However, we can confirm that we identified gaps in care delivery of peri-operative care as perceived through the eyes of the patient. We did not explore the predictors of gaps.

  1. From the scientific point of view, it is more a study to identify the perceived quality of care of patients undergoing colorectal surgery, to look at the level of anxiety, the perceived care and the quality of these, to review there are validated questionnaires, which allow us to evaluate this perception, and the latest systematic reviews of it.

There are many patient-reported outcome measures (PROMS) that measure the patient’s health condition (such as anxiety, depression, quality of life and other aspects of well-being). However, we conducted a comprehensive review of the literature, and found there are no patient reported experience measure (PREMS) that explore patient’s views and observations about a comprehensive range of components of perioperative care delivery based on ERAS pathway.  (we added this detail to line 130)

  1. For future studies, they could assess qualitative studies that allow us to identify what these perceived needs by patients could be, and accompany it with quantitative studies as they present us, improving the variables that are intended to be investigated, the tools that are used, and better justify why, and what is intended to improve, in this case the gaps in information and the quality of care perceived by the patient.

We strongly agree that it is important to review existing qualitative and quantitative literature.  Indeed, our study-specific PREM survey was developed after extensively consulting the existing literature including systematic reviews that summarised both qualitative and quantitative methods (please see reference 26).

We have included this as a recommendation for future research (Line 364)

  1. I think you should do a review again and make a new article proposal, which improves the quality and viability of cross-sectional studies and of interest to readers, and clarifies the stages and professionals who have to carry out the interventions, let us know if it has happened an ethics committee, and better clarify the viability of the survey carried out and its dimensions in accordance with the objective or objectives set out in its study.

We agree a systematic review of patient reported experience measures in  perioperative care for colorectal surgery would make an important contribution to the literature, and added this recommendation for future research (added to line 372).

Please refer to the previous comments that have systematically addressed each of these other important concerns.

We hope our revised manuscript meets your approval.

Round 2

Reviewer 3 Report

Title:

I recommend removing the word Forecast from the title of the article, the title must respond to the stated objective of the study, adding a statement as a forecast, for readers it would not be correct to speak of provision, it is only advice, perception in care received and experiences perceived in preoperative care and information on colorectal cancer, cross-sectional study.

This study does not identify the ethics committee on the research project, nor conflicts of interest